# Models for Translational Proton Radiobiology—From Bench to Bedside and Back

**DOI:** 10.3390/cancers13164216

**Published:** 2021-08-22

**Authors:** Theresa Suckert, Sindi Nexhipi, Antje Dietrich, Robin Koch, Leoni A. Kunz-Schughart, Emanuel Bahn, Elke Beyreuther

**Affiliations:** 1OncoRay—National Center for Radiation Research in Oncology, Faculty of Medicine and University Hospital Carl Gustav Carus, Technische Universität Dresden, Helmholtz-Zentrum Dresden-Rossendorf, 01309 Dresden, Germany; theresa.suckert@uniklinikum-dresden.de (T.S.); sindi.nexhipi@uniklinikum-dresden.de (S.N.); Antje.Dietrich@uniklinikum-dresden.de (A.D.); leoni.kunz-schughart@oncoray.de (L.A.K.-S.); 2German Cancer Consortium (DKTK), Partner Site Dresden, and German Cancer Research Center (DKFZ), 69120 Heidelberg, Germany; 3Helmholtz-Zentrum Dresden-Rossendorf, Institute of Radiooncology-OncoRay, 01309 Dresden, Germany; 4Heidelberg Institute of Radiation Oncology (HIRO), 69120 Heidelberg, Germany; robin.koch@dkfz-heidelberg.de (R.K.); e.bahn@dkfz-heidelberg.de (E.B.); 5Department of Radiation Oncology, Heidelberg University Hospital, 69120 Heidelberg, Germany; 6National Center for Tumor Diseases (NCT), 69120 Heidelberg, Germany; 7National Center for Tumor Diseases (NCT), Partner Site Dresden, 01307 Dresden, Germany; 8German Cancer Research Center (DKFZ), Clinical Cooperation Unit Radiation Oncology, 69120 Heidelberg, Germany; 9Helmholtz-Zentrum Dresden—Rossendorf, Institute of Radiation Physics, 01328 Dresden, Germany

**Keywords:** proton therapy, proton RBE, toxicity, preclinical models, cell culture, organoids, tissue slice culture, mouse model, zebrafish, in silico modeling

## Abstract

**Simple Summary:**

An increasing number of cancer patients are treated with proton therapy. Nevertheless, there are still open questions that require preclinical studies, for example, those regarding long-term side effects or new treatment approaches. In this review, we discuss the main research topics of proton radiobiology and describe the typical challenges related to preclinical experiments in this field. We provide a summary of the different available preclinical models, and how they were applied to conduct proton-specific research in the past. This includes cell culture models of increasing complexity, animal studies, and computer simulations. In addition, we give an overview of possible endpoints and suggest models from other disciplines for adaptation to biomedical proton research. In doing so, we contribute to designing meaningful research studies in the future, which will ultimately help to improve patient treatment.

**Abstract:**

The number of proton therapy centers worldwide are increasing steadily, with more than two million cancer patients treated so far. Despite this development, pending questions on proton radiobiology still call for basic and translational preclinical research. Open issues are the on-going discussion on an energy-dependent varying proton RBE (relative biological effectiveness), a better characterization of normal tissue side effects and combination treatments with drugs originally developed for photon therapy. At the same time, novel possibilities arise, such as radioimmunotherapy, and new proton therapy schemata, such as FLASH irradiation and proton mini-beams. The study of those aspects demands for radiobiological models at different stages along the translational chain, allowing the investigation of mechanisms from the molecular level to whole organisms. Focusing on the challenges and specifics of proton research, this review summarizes the different available models, ranging from in vitro systems to animal studies of increasing complexity as well as complementing in silico approaches.

## 1. Introduction

Proton therapy has an inverse depth dose profile when compared to conventional radiotherapy with photons, which offers the possibility to reduce the dose delivered to the tumor-surrounding normal tissue. In long-term survivors, this dosimetric benefit can increase the quality of life [1] by decreasing the risk of normal tissue side effects [2,3,4]. For this reason, the number of proton beam facilities has been steadily rising in the last decade, with several new ones planned or already under construction [5]. To date, more than two million cancer patients have been treated with proton therapy [5]. Nevertheless, numerous open questions on basic radiobiology, physical effects, novel therapeutic strategies, and technical innovations have remained unanswered, calling for intensified efforts in translational research.

A topic of ongoing discussion is the proton relative biological effectiveness (RBE), which is assumed to be a constant factor of 1.1 in the clinical context. However, abundant in vitro, rare in vivo (summarized in [6]), and first clinical [7,8,9,10] data indicate a varying RBE, often with a distinct elevation at the distal end of the proton range. Several concepts exist for RBE-adapted treatment planning in clinical practice [11,12,13], but there is a considerable variation within the experimental data. RBE values strongly depend on different physical (dose, dose rate, linear energy transfer (LET), fractionation) and biological (tissue, model, endpoint) factors [13,14,15]. These issues call for caution regarding clinical implementation of a varying RBE and create the need for more preclinical studies, using sophisticated in vitro systems and in vivo models [13,16].

The suspected variable RBE, and other factors, such as proton range uncertainties or anatomical changes during treatment [17], can result in dose deviations and consequential normal tissue side effects in some patients. Moreover, organ-specific radiosensitivities may differ from clinical guidelines. For example, there are indications of an increased radiosensitivity in the periventricular region, leading to a higher incidence of radiation-induced brain injury [10,18]. Out-of-field effects have also been observed, especially after irradiation of larger volumes [19]. Thus, these normal tissue toxicities require further research in order to adapt clinical guidelines for optimal patient outcomes.

While normal tissue sparing is the main advantage of proton therapy, the radiation response of the tumor has to be considered as well. Differences between photon and proton irradiation in deoxyribonucleic acid (DNA) damage induction and repair are still being unraveled [20]. In addition, there are indications that particle irradiation not only influences cancer cell migration and invasion [21,22,23], but also does so differently from photon irradiation [24]. These differential responses may influence treatment outcomes, especially in chemoradiotherapy settings. Furthermore, new opportunities for combinatorial targeted therapies or radioimmunotherapy [25] are emerging that require experimental validation. In the field of proton therapy, research activities not only focus on biological mechanisms and new clinical strategies, but also need to aim for technical improvements and novelties. Examples for the latter are innovative beam delivery concepts, such as FLASH irradiation [26,27] or proton minibeams [28,29], both of which promise increased normal tissue protection. However, the mode of action of these modalities and their optimal treatment parameters are not yet fully understood and require systematic studies.

The above-mentioned research questions cover a large variety of topics; yet, their common denominator is the requirement of suitable preclinical models to complement or precede clinical trials. The preclinical setting enables the investigation of radiation effects on a molecular, cellular, and systemic level, which contributes to our understanding of the underlying mechanisms. In addition, aspects such as the safety and effectiveness of new treatment approaches can be tested before designing clinical studies. A prerequisite in translational research is clinical relevance. This does not mean that the most complex model and experimental setup, i.e., an in vivo study, is always necessary, but rather the one most suitable for the research question at hand. For example, patient-derived three-dimensional (3D) cell culture models can yield more significant results than animal experiments, and in silico methods can help to make efficient experimental design decisions in conventional biological studies. In this context, our review summarizes the existing preclinical models for proton therapy research, highlights examples for their application, and offers conclusions on useful experimental endpoints. We particularly emphasize restrictions and challenges in the field of proton radiobiology and discuss how they can be overcome to gain the most relevant insights for clinical implementation.

## 2. Characteristic Challenges in Preclinical Proton Research

### 2.1. The Physicist’s Point of View

The performance of proton experiments faces several challenges that might influence the choice of model and endpoint. Access to proton facilities, one of the main bottlenecks in former times, has clearly improved in the last years with newly operating proton therapy facilities that include dedicated experimental areas, e.g., [30,31], allowing for radiobiology and physics experiments in parallel to patient treatment. Focusing on collaborations and scientific exchange, the Inspire project of the European Union provides a network and transnational access program between European proton facilities, clinical and research ones, that enable radiobiology experiments [32]. Clinical proton centers are organized under the umbrellas of the globally active Particle Therapy Co-Operative Group (PTCOG) [5] and the European Particle Therapy Network (EPTN).

Currently, systematic proton studies face two main challenges: (1) the precise and reproducible positioning of samples, and (2) accurate absolute dosimetry, especially for small volumes and at the distal edge. These two issues cannot be considered independently since sample positioning along the proton depth dose curve defines the necessary corrections for beam quality [33] and LET [34], which have to be taken into account by dosimetry. Thereby, preclinical dosimetry refers, in principle, also to clinical dosimetry standards but differs in details, such as the composition and dimensions of materials in the beam path [35] and the size of the target volumes. For adherent cellular monolayers, which are typically only a few microns thick, positioning is straightforward and can be realized with high accuracy using water phantoms [30] or water equivalent material [36,37] for range compensation. In combination with dose simulations, these approaches allow, for example, the determination of cell survival data dependent on proton LET [36,37]. The irradiation of 3D cultures is more complicated, especially at horizontal beam lines that demand for the upright positioning of samples. Spheroids, for example, roll down the agarose bed if the culture vessel is tilted to $90^{\circ }$; this has to be taken into account for positioning and dosimetry. Organ slices and 3D cultures in gelatinous matrices cannot be irradiated upright and should, therefore, be investigated at vertical proton beams [38] or require dedicated irradiation setups [39,40].

Unlike in vitro studies, in vivo ones include the irradiation of different volumes—from whole organs [41] and extended tumor volumes to subvolumes of organs and small orthotopic tumor models—depending on the research question. Here, precise and reproducible positioning is key, especially for 3D targets inside the animal body. Elaborated positioning for animal experiments is realized at the different proton centers [31,42,43,44,45,46,47,48], which is, however, difficult to standardize. In a first attempt, Gerlach et al. developed a portable setup for animal studies with protons [46]; some facilities installed small animal irradiators, enabling in situ CT imaging and precise positioning of target volumes [49,50]. Besides positioning, small animals are also challenging for absolute dosimetry, due to a lack of standardized dosimeters for such volumes. Practical solutions are small dosimeters, such as alanin pellets, and radiochromic films that can be cut into user-defined shapes and sizes [51,52,53]. Experiments with new radiation qualities, such as FLASH radiotherapy, laser-driven sources or proton mini-beams, demand for adapted solutions with respect to sample positioning and dosimetry [54,55,56,57]. Moreover, point-like measurements with small dosimeters can be supported by simulations [58] to resolve proton dose distributions [59,60]. Standardized 3D phantoms of rodents could clearly improve preclinical proton dosimetry, but are currently only available for orthovoltage X-rays [61].

On a final note, it should be mentioned that the correct reporting of all these physical beam and proton field parameters has acquired an increased importance during the last years [62,63,64]. Several expert groups have released recommendations on profound reporting for preclinical proton experiments, for example, the ESTRO-Advisory Committee for Radiation Oncology Practice [65].

### 2.2. The Biologist’s Point of View

In recent years, medical research had to cope with the so-called “reproducibility crisis”: scientists report that they are not able to reproduce experiments [66], and breakthroughs from preclinical research fail to deliver the hoped-for clinical impact. For example, many promising novel cancer drugs that had proven effective in preclinical studies failed in clinical trials [64,67]. Considerable contributors to the problem are poor methods reporting and low statistical power of experimental designs [66]. Another aspect is that the chosen experimental model(s) and readouts often neither adequately reflect the selected clinically relevant endpoints nor represent the clinical reality [67]. Accordingly, state-of-the-art preclinical proton therapy testing needs well-defined experimental settings to avoid a similar waste of time and resources and to not slow down the overall progress needed to improve patient treatment.

The requirements for optimal preclinical models are high. First and foremost, the transferability to patients needs to be ensured as far as possible. Other essential factors are financial affordability, reasonable throughput, reproducibility, and the availability of relevant experimental endpoints. While all preclinical studies have to deal with these restrictions, proton radiobiology is additionally challenged by the assignment of beam time. At both clinical and experimental proton accelerators, the available beam time for radiobiological experiments is limited, which demands meticulous preparations, including the choice of suitable models and readouts. If cells or tumor models are not growing at the anticipated speed or if a technical failure occurs, simple postponing of the experiment is often not possible. Hence, physicists and biologists need to work hand in hand in proton radiobiology to optimize the experiments for maximum output. This includes not only careful preparation of and support during the experiment, but also the comprehensive in silico description of the experimental setup, model and results. What appears to be challenging and work-intense is at the same time beneficial since the close interdisciplinary collaboration opens completely new and innovative research approaches. For example, radiobiological data can be described and simulated very well under physical aspects; thus, in silico experiments can contribute vastly to our current understanding.

The following chapters describe the available models and respective endpoints applicable in proton radiobiology along the translational chain from two-dimensional (2D) and 3D cell culture to small animals, large mammals and in silico concepts (Figure 1) The advantages and disadvantages of each approach are briefly discussed, along with examples of its successful application and recommendations for optimal use. For clarity and synopsis, the respective models, as well as their specifics and a few representative endpoints are outlined at the beginning of each chapter.

## 3. In Vitro Cell Culture Models

This chapter provides an overview of the different cell culture models that were currently or might in future be used in proton radiobiology. As summarized in the graphical outline (Figure 2), the models are presented from the less complex 2D cell culture to spheroids, organoids and more complex 3D models.

### 3.1. 2D Cell Culture

The 2D cell culture is an extensively used in vitro model. Both primary and established cell lines are simple and economical tools used to investigate various aspects of radiobiology. Primary cell lines are more reflective of the in vivo genetic features, as they are isolated directly from human or animal tissue [73]. Established cell lines, however, are applied more frequently because of their infinite lifespan, known characteristics, standardized culturing, and ease of genetic manipulations. Moreover, the usage of established cell lines circumvents ethical concerns associated with the use of animal and human specimens [74]. In radiotherapy research, cell survival is a standard biological endpoint quantified by the colony formation assay (clonogenic survival assay). Other endpoints, such as apoptosis [75,76], reactive oxygen species [76], chromosome aberrations [77], the quantification of DNA damage and repair proteins, as well as the resulting gene expression changes [20,78,79,80,81,82], are studied to obtain a more detailed insight into radiation response mechanisms [14].

Established 2D cell lines play a pivotal role in the radiobiological characterization of proton irradiation. They are applied for comparing different proton facilities [83] and the evaluation of new treatment modalities, such as proton monoenergetic arc therapy [84], intensity-modulated proton therapy [85,86], spot-scanning proton therapy [87] and FLASH irradiation [78]. Well-known cell lines in this respect are V79 [88,89], derived from hamster lung fibroblasts, as well as human cell lines, such as H460 (large-cell lung carcinoma), HSG (human salivary gland tumor) [83,90], and normal human fibroblasts [78,91], which are more relevant for patient treatment. Once established at a facility, these cell lines are frequently applied to study proton related effects, such as RBE [14], oxygen enhancement ratio [90], and LET. Generally, 2D cell cultures are the preferred model in LET studies since adherent cell monolayers of a few microns in thickness allow for a high positioning accuracy [36,37], unattainable in 3D cell culture or animal models [45]. For this purpose, various murine and human tumor [20,37,92], and normal tissue cell lines are used (summarized in [14]). The effect of LET and RBE in fractionated proton exposure has been studied in human fibroblasts [36]; however, the fractionation effects are still scarcely researched.

Additionally, 2D cell cultures are appropriate tools to investigate the radiation response of cells to different treatment modalities. Several studies have focused on the effect of proton irradiation on disease mechanisms, such as tumor invasion and migration for skin [22,24], brain [21], lung [93], and breast cancer [94,95,96,97]. Others have investigated how cells respond to different drugs, which is of the utmost importance in (proton) radiooncology, where chemoradiotherapy is a standard therapy approach. In clinical practice, treatment protocols of photon-concurrent chemotherapy are simply transferred to proton therapy, neglecting potential differences between these approaches [98]. As such, 2D cell culture can test the radiosensitizing effects of available chemotherapeutic agents [99,100] and other drugs [101] in combination with proton irradiation on a range of cancer types. Screening platforms of well-known human tumor cell lines with heterogeneous genomes that mimic inter-patient variability are used to identify chemoradiotherapy susceptibility after photon irradiation [102,103] and should be extended to translational proton therapy approaches. Prospective markers and radiosensitivity genes can also be validated and studied in genetically modified cell lines in order to investigate radiation responses. As an example, a study showed that FANCD2 knockdown cell lines were more sensitive to proton rather than X-ray irradiation [104].

The advantages and wide applications of established 2D cell lines should be considered on a study-specific level. When chosen as a model, attention needs to be paid to the fact that they grow in a monolayer and, therefore, cannot mimic the complex in vivo tissue architecture and microenvironment. This hinders interpretation of both the tumor and normal tissue results for which therapy responses may depend on the vascularization and the 3D structure that provides different access to nutrients, oxygen, and therapeutic agents. Moreover, elaborated biological mechanisms, such as the progression of organ-specific early and late effects [14], demand more complex models.

### 3.2. Spheroids and Organoids

The 3R principles, i.e., replacement, reduction and refinement of animal experiments [105], continue to gain in importance across all scientific disciplines, including translational research, which has led to an increased interest in intricate and relevant in vitro models. In this regard, it is beyond dispute that 3D cultures are more realistic and informative than 2D cell systems [106,107,108,109]. The most straightforward approach in radiobiology is the 3D clonogenic survival assay, in which single cells grow into 3D cell clusters when cultured in extracellular matrix components [107,110,111]. More complex 3D structured models, such as multicellular spheroids, further reestablish histomorphological, pathophysiological, and microenvironmental features that better resemble the in vivo situation [106,109,112,113]. Their main characteristics depend on the size and can include 3D cell–cell and cell–matrix interactions, radial gradients of oxygen, nutrients, pH, catabolites, cellular proliferative activity, and in vivo-like differentiation patterns; they can develop therapeutically-relevant hypoxic regions and a secondary necrotic core [106,113,114,115,116,117]. Another level of complexity is achieved by establishing patient-derived organoids from normal and tumor tissues of various entities [118,119,120,121,122,123]. They are considered the culture models closest to the individual patient, mirroring the heterogeneity and genetic background of the original tissue [124,125]. Organoids can also be grown from mouse tissue, which opens a wide range of possibilities for biological studies, including the use of transgenic donors [126,127]. However, some research questions call for even higher complexity mirrored by the co-culturing of multicellular spheroids or organoids with various stromal cell types, such as fibroblasts, endothelial, or hematopoietic cells [128,129,130,131].

From the radiobiological perspective, 3D cell cultures have already proven to be the most valuable tools. The work with 3D models has contributed to understanding the altered responsiveness of chronically hypoxic tumor cells, and the role of cell–cell and cell–matrix interactions in radioresistance [106,109,112,116,132]. The 3D clonogenic survival assay was recently applied for RBE studies with protons and carbon ions [40,111], and to evaluate the outcome of novel molecular targeted agents in combinatorial treatments for pancreatic and HNSCC cell line models [40,133]. A review from Walenta and Mueller-Klieser [134] summarizes the experimental studies from 2D and 3D cultures dedicated to evaluating the RBE and side effect mechanisms of heavy-charged particles. Multicellular spheroids were also applied to study proton irradiation alone [135] and in combination with chemotherapy [136,137], to compare the RBE of spot scanning and passive scattering beams [138], and to validate the biological effectiveness of proton FLASH irradiation [139]. Furthermore, Brack et al. recently proved the technical feasibility of their irradiation device with laser-driven particles by visualizing DNA damage in a spheroid model [72]. Organoids derived from stem or progenitor cells are of utmost relevance for investigating normal tissue side effects, as they allow for stem cell-related response studies [125]. Normal tissue organoids, originating from, for example, the intestinal system, the salivary glands, or the mammary glands, were used to assess tissue radiation sensitivity and irradiation-induced toxicity mechanisms [127,140,141,142]. These models should be exploited in more detail to shed further light on the biological consequences of proton irradiation in normal tissues. In this context, it was shown that a magnetic field does not modify the response to proton irradiation in stem cell-derived salivary gland organoids [143]. Another impressive example is the study of Nowrouzi et al., who evaluated the gastro-intestinal response to photon, proton, and carbon ion radiation using transcriptome profiling of irradiated patient-derived human intestinal organoids [38]. Nowadays, studies in the field of tumor organoid-based personalized medicine more frequently consider radiation therapy (RT) [123,144,145,146]. Although trials using cancer-derived organoids treated with particle therapies are still lacking, they will soon inform decisions in the field. As proton therapy cannot be offered to all patients, there is a need for stratification. Here, patient-derived organoids represent a powerful tool for individualized treatment decisions [122,128]. Future trials should include pairs of normal and cancer tissue-derived organoids from the same patient to evaluate the full range of individual therapeutic windows [125,128].

Notably, the experimental design and analytical endpoints used in 3D culture assays can critically differ, thereby defining the grade of in vivo resemblance. Most 3D clonogenic assays are based on the irradiation of single cells [110,111,147], which is still artificial because direct cell–cell interactions and some radiotherapeutically relevant (patho)physiological phenomena are not present during exposure. On the other hand, the assessment of clonogenic survival of irradiated cell clusters, spheroids, or organoids requires dissociation of the cultures upon treatments and subsequent seeding of single-cell suspensions. What appears to be a straightforward approach can become quite vulnerable to artifacts because different treatments may affect the cells’ susceptibility to enzymatic and mechanical dissociation stress. Alternatively, cell viability assays adapted from classical drug response assays [135,148] can be applied to monitor the metabolic activity of 3D cultures after treatment, as is often used in clinical trials with organoid cultures [123,144,145]. However, the results critically depend on the time of measurement and may not be suitable for all types or sizes of 3D cultures. Moreover, such endpoints do not reflect clonogenic survival, which is still considered one of the most important in vitro readouts in radiobiology. One promising method for assessing the treatment outcome remains image-based monitoring of 3D cultures over time. It allows visualizing culture integrity and subsequently determining volume growth kinetics, e.g., in spheroid growth delay studies [115,137,149] or patient-derived organoids [144]. The state-of-the-art spheroid control probability assay represents a clinically relevant endpoint for experimental radiotherapy [149,150,151,152], analogous to the tumor control probability (TCP) and tumor control dose 50% (TCD50) assessment in vivo [153]. Such analytical endpoints are essential for curative treatments and show great potential for systematic proton irradiation studies [136]. Furthermore, radiobiologists are encouraged to adopt some assays from sphere and spheroid cultures to organoid cultures. However, ongoing efforts are required to translate these long-term outcomes into clinically relevant endpoints that can be assessed more rapidly in organoids for screening purposes and personalized (proton) radiotherapy, for example, based on DNA damage analysis for assessing radiation sensitivity [111,154,155].

### 3.3. Thin-Cut Tissue Slices and Other 3D Cell Culture Models

Another cell culture model offering 3D architecture is thin-cut tissue slices, which grow on specialized inserts at the interface of the medium and air. Both tumors and normal tissues can be cultured in this fashion. The former derive from either tumor-bearing animals or patients using surgical resections, whereas the latter mainly stem from rodents. The most widely used normal tissue in slice culture is the neonatal brain [156] but others, such as lung [157] or heart [158] slice cultures, exist as well.

Tissue slices offer several advantages compared to other cell culture systems, such as realistic heterogeneity, preserved tissue morphology, and high success rate during culture generation [159]. Therefore, they have been applied in RBE investigations [160], pharmacodynamic profiling [161], the testing of novel treatment compounds [162], and comparing different chemoradiotherapy combinations [163]. In addition, researchers examined the tumor microenvironment and cell invasion processes in tissue slices. Both thin-cut tumor slices alone [164] and co-culture systems of tumor cells and organotypic slices [165,166] were applied for such studies. Tumor slice culture in particular is considered a suitable tool for personalized medicine, enabling the comparison of treatment approaches prior to therapy for optimized patient outcome [162,163,164,167]. Relevant radiobiological endpoints for thin-cut tissue slices include the analysis of apoptosis, [160,163], proliferation [168], and DNA damage [39,162,168]. In addition, some functional assays exist that measure metabolism or cell death in slice cultures [39,167,169]. However, in contrast to spheroids and organoids, no data on tumor control probability rates can be achieved with this model. Other drawbacks are a limited culturing time, missing vasculature, as well as undesired functional and transcriptional changes during culturing [159,170]. So far, only a few studies have applied thin-cut slices as ex vivo platform for particle therapy with protons [39] or heavier ions [160,163,168]. Nevertheless, all above-mentioned experiments can be easily adapted for preclinical proton experiments; with the growing need for implementing the 3R principles, their number will likely increase.

While most radiobiology research has been performed on simpler 3D models such as spheroids, some studies are applying complex tissue-engineered approaches, such as 3D scaffolds or organs-on-a-chip. These models are often commercially available, thus potentially offering higher reproducibility across laboratories. Strikingly, particle experiments so far have all focused on different skin models, which were used to investigate proton RBE [171], normal tissue side effects of proton microchannel therapy [172], and LET-effects of carbon irradiation [173]. Despite promising results, these models are still rarely applied in preclinical proton research due to their high costs. With more suppliers on the market, this may change in the future.

On a final note, organ-on-a-chip applications may offer solutions to many above-mentioned drawbacks of other cell culture models. They enable the investigation of both tumor and normal tissues, including the microenvironment, cell–cell interactions, and even organ functions [174]. Unfortunately, data on particle irradiation of organ-on-a-chip systems are not available yet, but first radiobiological applications are promising [175,176].

## 4. In Vivo Models

This chapter summarize the in vivo models that were applied in proton research from the small teleost vertebrates, to rodents and rabbits, and finally larger mammals like cats, dogs, pigs and apes. Common specifics of these models as well as some representative endpoints are shown in Figure 3.

### 4.1. Teleosts

During the last years, the interest in zebrafish (*Danio rerio*) and medaka (*Oryzias latipes*) as small animal models has steadily increased, also for radiobiological research [177,178]. Both teleosts produce embryos in a transparent chorion, enabling easy detection of morphological malformations [68,179] by light microscopy. Extracorporeal embryonic development and whole genome sequencing facilitates genetic manipulation [177,180], compared to mammals. Moreover, their fast development and a high number of embryos per breeding pair make teleosts attractive for systematic studies, e.g., those on radiation effects [68,181,182] or combined treatment modalities [179,183]. Regarding proton research, the small size of the embryos of 1 mm in diameter one day after fertilization allows for irradiation in cell culture vessels and accurate positioning [68,181]. For example, Szabo et al. [68] used zebrafish embryos to determine the RBE of entrance and spread-out Bragg peak protons relative to a 6 MV photon beam, whereas Li et al. [181] evaluated the response to 8 MeV protons. More recently, zebrafish embryos became attractive to evaluate new proton radiation modalities, such as FLASH irradiation [184]. Adaptive response to proton microbeams [185] and altered blood vessel formation after proton irradiation [186] are two endpoints studied in adult zebrafish.

One step further, the injection of tumor cells enables the real-time and visual observation of tumor cell metastasis in zebrafish embryos [187,188]. The successful implementation of gastric [189] and colorectal [187] patient-derived xenograft (PDX) tumors resulted in the idea of using zebrafish as “avatars” for personalized medicine [190,191]. However, besides all advantages and possibilities, one should not forget that fish and mammals differ in many points, which might also affect radiation susceptibility and the treatment response. Following the translational chain, findings in teleosts should always be verified in mammals.

### 4.2. Rodents

Murine models, i.e., mice and rats, are conventionally considered the final link in the translational process to move forward from preclinical findings to clinical trials. Their advantages over large animal models are the small size, which facilities housing, lower economical expenses, as well as short reproductive cycles and lifespans. Additionally, genetically engineered mice can be utilized to investigate the relevance of specific genes, and their responses and changes to radiation [192,193]. In radiobiology, murine models are typically used to study the elaborated mechanisms of radiation-induced normal tissue and tumor response.

An important parameter affecting proton radiation-induced side effects is the RBE, which has been investigated in a number of in vivo experiments on mouse [41,47,193,194,195,196,197] and rat models [45,198,199,200,201,202,203] to estimate its values and dependence on other factors, such as LET, radiation dose and fractionation regime. The biological endpoint chosen for the RBE estimation highly depends on the tissue type. For spinal cord studies, myelopathies, such as paresis [45,199,201] have been analyzed in rats, which develop radiation-induced symptoms similar to humans [204,205]. The intestine crypt survival assay [206] is a standardized method used in murine models to measure the gastrointestinal toxicity, a major dose-limiting factor during abdomen or pelvic irradiation. This approach is suitable to understand the differences between photon- and proton-induced intestinal injury [207] and is often used to compare proton facilities worldwide [193,208,209,210,211,212].

Another strength of in vivo models is reverse translation, i.e., replicating clinical observations in a preclinical setting. For example, recent studies have established proton irradiation of mouse brain subvolumes [42,59] to investigate the underlying causes of radiation-induced brain lesions appearing after proton therapy [10,18]. Overall, central nervous system (CNS) toxicity studies in rodent models are of increasing interest and performed from both a behavioral [213,214,215] and histological perspective [59,216,217]. Using clinically relevant settings is crucial in developing protocols that reduce normal tissue complications. Amongst others, this includes the development of proton mini-beam therapy [28,29] and proton FLASH irradiation [50,57,218,219]. Further studies on normal tissue toxicities have focused on processes, such as the peripheral inflammatory response [220], radiation-induced thoracic injuries [19,221], radiation-induced abdominal injury [207,222], as well as tumor incidence after irradiation of a healthy brain [223] and dorsal skin [224].

While sparing damage to the normal tissue is a critical aspect of RT, its primary goal is successful tumor eradication. Evaluation of radiotherapy treatments is often performed in rodent tumor models, using assays such as the TCD50 and tumor growth delay [153]. Orthotopic tumors are more clinically relevant, as the tumor is injected at the site of origin, but they also call for more complex irradiation protocols than subcutaneous ones. One example from the field of proton radiobiology is the study from Kwon et al. [225], which investigated the effect of proton therapy on tumor invasion and metastasis in the murine 4T1 breast tumor model. Orthotopic transplantation has also been applied for brain tumor studies in a number of syngeneic rat models, for example, to compare conventional and proton minibeam RT [226,227,228]. Allograft C6 brain tumor-bearing rats served as a model to evaluate the feasibility of gene therapy together with proton radiation as an innovative approach with the potential to enhance the outcome of radiotherapy [229]. The majority of orthotopic tumors are syngeneic or allograft tumors, making them also suitable for immunological studies. This is, however, accompanied by the disadvantage of using animal cell lines, and thus, findings have to be considered with caution before they are extrapolated to human cancer therapies.

The similarities between animal tumor models and patients are higher in xenograft models. These tumors are usually inoculated subcutaneously to overcome technical limitations that come with orthotopical transplantation [230] and irradiation. In addition, they better depict the underlying biology and response of human cancers to radiation alone or in combination with different drugs. Xenograft tumor models have been used to investigate the effectiveness of proton irradiation for head and neck squamous cell carcinoma [231] and triple-negative breast cancer [232], as well as proton irradiation with ultra-high dose rates [233]. While studies that evaluate combination therapies with proton irradiation are still scarce, the cell-derived xenograft model is valuable for this research question. For example, Waissi et al. [234] found that application of gemcitabine- and olaparib-based chemoradiotherapy in such models displays a higher effectiveness when using proton therapy. Another preclinical study deduced that the combination of proton beam therapy with targeted radionuclide therapy can produce a type-dependent additive or synergetic effect [235]. Meanwhile, data on radioimmunotherapy with protons are still missing, but mouse xenograft studies using other radiation modalities have shown promising results [236,237].

In contrast to the cell-derived xenograft model, PDX are considered a suitable model for personalized oncology. They appear to have a higher predictive power, in particular for individual clinical outcomes. PDX are used to study tumor characteristics, develop metabolic and imaging biomarkers, facilitate clinical trial design, as well as prioritize therapeutic targets on a patient-based level [238]. A drawback of the xenograft models is that the tumor microenvironment can still deviate strongly from clinically relevant settings, depending on the tumor entity [239]. Another disadvantage is the suppressed immune system of the mice, which is necessary to prevent graft-versus-host disease after xenotransplantation. One solution to this would be humanized mice [239], but currently the complexity of this model is hardly compatible with the limitations of proton research, especially regarding the cost–benefit ratio.

Naturally, murine models offer the possibility to investigate the in vivo therapeutic window of radiation, which is defined by both tumor and normal tissue reactions. This research is especially interesting for preclinical studies with proton FLASH irradiation, which promises normal tissue protection at unvarying tumor control rates. Indeed, two recent studies could prove this effect in C57BL/6 mice [218,219]. Moreover, this valuable preclinical research has already culminated in the first feasibility study of proton FLASH irradiation in patients [240].

A less frequently applied rodent model for evaluating radiation responses is the Syrian hamster (*Mesocricetus auratus*) [241], which has not yet been used in particle therapy studies. Nevertheless, it has a rising importance in infectious disease research, which drives the development of laboratory procedures for this species [242]. In addition, the emergence of immunodeficient hamsters for xenograft studies [243,244] may increase the model’s relevance for cancer radiobiology in the future.

In conclusion, rodent models provide an important preclinical setting to test a variety of radiobiological questions on both tumor and normal tissues. While mice provide more methodological options, e.g., through gene manipulation, rats can be irradiated more precisely and show higher similarities to human radiation responses.

### 4.3. Rabbits

In proton research, rabbits are less frequently used to study normal tissue toxicity [245,246] and more so to investigate the cancerous tissue response to irradiation. The VX2 tumor model is an anaplastic squamous cell carcinoma induced in rabbits that became a standard tool [247,248] in oncology to study a number of solid human cancers [249]. VX2 tumor-bearing rabbits have been used to investigate proton irradiation effects for a multitude of entities with superficial tumors in the rabbit ear [250,251], as well as deep-laying ones in the lung [252,253], uterus [254], and pelvis [255]. The VX2 model is especially preferred when replicating and studying radiation-induced lung injury, as it overcomes limitations of immune-compromised murine models pertaining imaging, host immunity, and pathological changes [253,256]. This is because the syngeneic VX2 rabbit lung tumor model is relatively large and has more similarities to humans in terms of airway anatomy, which highlights the value of this model in investigating lung disease pathophysiologies [257]. Nevertheless, the advantages of this model are accompanied by the drawback of VX2 tumor being a neoplasm of rabbit origin with a microRNA profile that has little commonality to human patient samples [258]. In addition, rabbits have larger space requirements and animal handling is more complex than for rodents. This has led to a decreased use of rabbits as a cancer research model in recent years.

### 4.4. Higher Mammals

Large mammals, such as pigs, dogs, cats, and primates, better resemble the human anatomy and physiology than small animal models. Due to their larger size, clinical diagnostic tools and treatment devices can be utilized in preclinical research, which is particularly interesting for some reverse-translated issues. The most relevant results should be expected from primate studies, due to the close similarities to humans on a genetic, anatomical, and behavioral level. Indeed, one of the first publications on proton radiotherapy mentions tests on young monkeys [259]. In the last century, proton irradiation of primates has mainly been conducted in the context of military or space research, where it provided valuable findings on normal tissue toxicities [260,261,262]. Today, only a few institutes use primates as a model for radiation side effects [263,264,265,266], and just a single recent study irradiated with protons in their experiments [267]. Overall, the primate model is unlikely to play a pivotal role in future proton radiobiology studies in most cancer research institutions, due to high costs, long follow-up times, considerable ethical concerns, and constant public scrutiny.

Pigs and dogs, two other large mammals, offer similar advantages to primates concerning organ sizes, the immune system, physiology, and even genetics. Accordingly, radiation responses were frequently investigated in these models, in particular to assess normal tissue side effects [268]. Proton radiation of pigs was so far only carried out in the context of space research [269], but the species has been successfully applied with other radiation modalities to study normal tissue side effects in brain [69,270] and skin [271]. Dogs, on the other hand, were already deployed in the early days of proton therapy, starting with a feasibility study of this irradiation source with canine pituitary gland and mammary cancer irradiation [259]. In the following years, beagles were also used for dosimetric and histological investigations of proton eye irradiation [272,273]. Classical studies on canines have declined in the last years; nevertheless, they have been used in proton research in a different context: the pet model for cancer treatment. Preclinical studies with domesticated animals that developed cancer naturally, e.g., cats and dogs [271,274,275,276], have the benefit that these tumors frequently show histopathological similarities to human tumors [277]. Pets often reach a high age and, thus, are likely afflicted by other age-related diseases as well. Due to differences in diet, life-style, and a living environment resembling the ones of their owners, pets better represent a heterogeneous patient population. In addition, this type of preclinical research is favorably perceived by the public [277], as opposed to conventional animal studies, which are regularly associated with a social stigma. Several cancer treatments have been tested in companion animals with success [271,275,278], proving the utility of the model. An example for the use of pets in preclinical proton research is the study of Mayer-Stankeová et al. [276], who tested the safety of proton spot scanning in canine tumors. However, the less homogeneous conditions in these settings have to be taken into account, i.e., a higher sample size is required for statistically sound analyses. Additional challenges of the pet model are unclear ethical guidelines, extended experimental time frames, and a lower prevalence of certain tumor entities [279].

In summary, large mammals offer various benefits, compared to other preclinical models from a translational perspective. Nonetheless, they are also associated with several limitations, such as high costs, low throughput, ethical concerns, and high experimental complexity—not unlike clinical studies, but with lower clinical relevance. Therefore, the pros and cons of performing a study on higher mammals instead of a clinical trial have to be carefully balanced, according to the research question at hand. Even then, implementing experiments with large animals may often not be technically feasible at existing beam lines due to missing infrastructures. Nevertheless, the option for irradiating large animals needs to be considered when designing and building new proton facilities with research units.

## 5. In Silico Models

In this chapter, a concise summary of the manifold in silico models, which are used to describe, understand and support proton radiobiology, is given. The list might not be exhaustive, but contains models on different dimensions and complexities, as briefly summarized in Figure 4.

Computational models are crucial tools in proton radiobiological research due to the inherently quantitative nature of biological dose responses to radiation. While analytical mathematical models have been used in radiobiology since the beginning, today, stochastic numerical methods are employed more and more frequently. Due to the high complexity of the studied biological systems, a purely mechanistic description is still not achievable. Most models are therefore purely phenomenological or semi-mechanistic. Between the different scales of interest from molecules to entire organs, models vary substantially in methodology, complexity and predictive power. An extensive review on radiation response models can also be found in [282].

### 5.1. From Particle Tracks to DNA Damage

A substantial portion of radiobiological research is concerned with the question how biological damage is quantitatively related to radiation characteristics. Radiation damage to cells has been shown experimentally to correlate with the amount of double-strand breaks. This amount depends not only on the (macroscopically averaged quantity) dose, but more precisely on the distributions of energy deposition events at molecular scales, as described by track structures. These are commonly modeled stochastically, for which a large number of Monte Carlo (MC) particle transport codes exists [280,283,284,285,286,287,288]. The different codes are discussed in detail in [289]. Since the required computational efforts limit these calculations to microscopic scales, MC codes apply approximative calculations for macroscopic simulations of dose and LET distributions, using the condensed history approach [290,291].

The calculated microscopic distributions of energy deposition events can then be used to model distributions of DNA damage. Here, the diffusion and reaction of chemical radicals also play important roles regarding indirect damage, which requires additional modeling [292]. In a next step, DNA repair mechanisms are often simulated with systems of differential equations [293,294,295] or MC approaches [296,297,298,299]. Such modeling approaches allow gaining insights into, for example, the effect of chromatin geometry [300,301], or the relation between LET and radiation modality [302,303,304] on the RBE of different DNA damage endpoints.

### 5.2. Cellular Scale

The relation between radiation and cell response is highly complex and depends on various signaling cascades. In response to radiation-induced DNA damage, cells can undergo a variety of responses, such as cell cycle arrest, apoptosis, or mitotic death. In vitro, the cell-level dose-response is commonly assayed by scoring cell survival, i.e., the proportion of colony-forming cells, using the clonogenic survival assay. The observation that cell survival does not decrease exponentially with dose, but forms a “shoulder”, has led to a multitude of models attempting to provide mechanistic descriptions of the observed behavior. Since the 1980s, the phenomenological linear-quadratic (LQ) model (reviewed in [281]) has been well established for all practical purposes, where the logarithm of cell survival as a function of dose is described by a second-order polynomial fit. Early mechanistic models attempted to explain the observed behavior by assuming that lethal lesions to a cell (such as asymmetric exchange-type aberrations) may occur either directly or indirectly via some second-order rate process, such as the interaction of two DNA double-strand breaks [305,306]. Later models assumed that lesions may or may not develop into lethal lesions, depending on the success and kinetics of DNA repair processes [307,308,309]. A generally accepted explanation is still lacking (reviewed in [282,310]). Of note, all mentioned models rely on experimental cell survival data as scored in the clonogenic survival assay. Recently, the validity of the assaying procedure was called into question after time-lapse imaging revealed a large heterogeneity of colony growth rates after irradiation [311].

Many computational RBE models (summarized in [312]) are similarly constructed as the above-mentioned models, with the difference that the occurrence of lethal and sub-lethal lesions depends, besides dose, on additional parameters describing beam physics (commonly LET). The semi-mechanistic local effect model and the microdosimetric kinetic model employ track-structure assumptions and are particularly used for carbon ion radiation [313,314]. For proton radiation, several established phenomenological models share a similar structure, where the LQ parameters are modeled as functions of LET [315,316,317]. The predictive power of these models is hampered by the fact that the underlying experimental data exhibit a large variability [14].

### 5.3. Cell Aggregate Scale

Spheroids, organoids and tumor masses are often modeled stochastically, using methods such as cellular automata, the Potts model or agent-based models, made up of individual cells as base units [318]. The effect of radiation on tumor masses has also been modeled stochastically by numerous models, frequently with a special focus on the effects of tumor vascularization and oxygenation (reviewed in [319]).

The intestinal crypt survival assay is an established endpoint that allows to study dose response on a cell level in vivo [206] and yields important data for the RBE of protons [14]. It is commonly modeled by simplified analytical formulas [206], which have been used extensively for generating biological hypotheses in radiobiological and stem cell research [320,321]. Novel stochastic modeling approaches of the biological processes as well as the assay procedure yield, for example, mechanistic insights into the dose–volume effect [322,323].

### 5.4. Tissue Scale

The goal of most radiobiological research is quantifying the effect of radiation on organs and tumors to improve the outcome in clinical applications. Yet, the mechanisms of these effects are poorly understood, and tissue-scale models are largely phenomenological. Tumor control is assumed to be achieved when all clonogenic tumor cells have been depleted by radiation, which forms the basis of most TCP models [324]. However, this paradigm has been increasingly challenged by new biological insights [325]. In normal tissues, the dose response differs largely from what would be expected from simple cell killing. The most important differences lie in the dose–volume effects, i.e., complex relations between spatial distributions of dose and outcome [199,326,327] and in the occurrence of late radiation effects with unexplained long latencies and complex dose–latency relations [326,328,329]. Normal tissue complication probability (NTCP) models aim to capture dose–volume effects, employing either empirical scaling laws [330,331], or dividing the tissue into hypothetical sub-units, which give rise to the concepts of serial and parallel organization [332,333,334]. Novel modeling approaches employ methods from statistical physics [335].

Since these observations are similar in both preclinical and clinical data, there is great interest in translational modeling [336] and several models are routinely used in clinical practice. In treatment planning, dose–volume effects are incorporated in the form of constraints and objectives of optimization algorithms (e.g., equivalent uniform dose and dose–volume histogram parameters). In carbon ion therapy, a model-based RBE is used for treatment planning [337,338]. In the Netherlands, NTCP model-based patient selection is being used to decide between photon and proton therapy [339]. For clinical data analysis, voxel-level modeling is increasingly employed to resolve the spatial information of complications, especially with regard to the RBE of proton radiation [8,10,340,341,342].

## 6. Conclusions

In line with the increasing number of proton facilities, proton RT has become part of the standard care for certain tumor types. However, despite the growing number of treated patients, further research is still needed to improve therapy and answer numerous open questions, e.g., those on potential differences in proton and photon radiobiology or alterations in chemoradiotherapy effects. In addition, new trends emerging in clinical oncology would benefit from preclinical research. For example, one recent treatment approach with great potential is combined immuno-radiotherapy, but unfortunately, no data with proton radiation are available so far; thus, translational efforts to combinatorial proton therapy are highly in need. Some of these points might be answered in clinical trials, which can nowadays be organized as multicentric studies, due to the increasing number of proton centers worldwide. Nevertheless, specific mechanistic or radiobiological aspects should be investigated in translational research studies (Figure 1) under careful consideration of models and endpoints. Starting from in silico approaches to large animals, each model described in this review has specific characteristics that might favor it over the others, depending on the research question at hand (Table 1). Mechanistic studies are most frequently performed in in vitro systems since cell cultures are easy to manipulate and different treatment modalities can be systematically tested in a short time. Other topics, such as tissue responses to radiation, demand for animal models, which need to be chosen, according to the available proton beam parameters and experimental equipment at each proton site.

Although the models highlighted in the present overview follow the standard translational research chain, it is worth noting that this chain is not a one-way street. Preclinical studies are often performed in forward translation to answer basic questions, for example, on the LET dependence of the proton RBE, translating the obtained results into clinical trials. However, clinical problems can also be back-translated to the laboratory to gain insight into the underlying mechanisms that cannot be studied in patients. Both forward and reverse translation in proton RT research are at the intersection of medicine, radiobiology and physics. Therefore, translational proton research requires interdisciplinary efforts, where physicists, biologists and clinicians share knowledge and skills to answer complex radiobiological scenarios with adequate preclinical models.

## Figures and Tables

**Figure 1 cancers-13-04216-f001:**
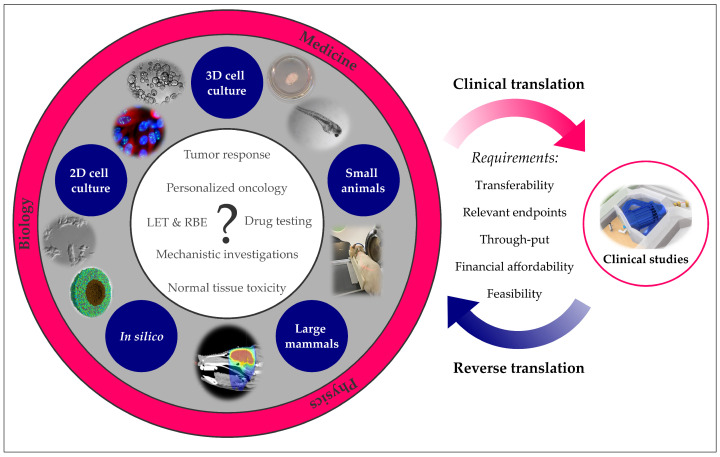
Translational research in (proton) radiobiology. Proton-specific research questions can be answered with a variety of preclinical models ranging from 2D cell culture to higher mammals. In silico approaches may deepen the understanding of underlying mechanisms. Preclinical insights can help to design clinical studies, and clinical observations can be back-translated into preclinical models. Images (clockwise): Tumor slice culture [39], irradiated zebrafish embryo [68], rat proton irradiation setup at Institut Curie [32], photon treatment plan of a mini-pig brain [69], simulated tumor spheroid 40 min post photon irradiation [70], migrating human uveal melanoma cells [22], fluorescence staining of HeLa cells [71], human pancreatic cancer organoids (courtesy of Max Naumann).

**Figure 2 cancers-13-04216-f002:**
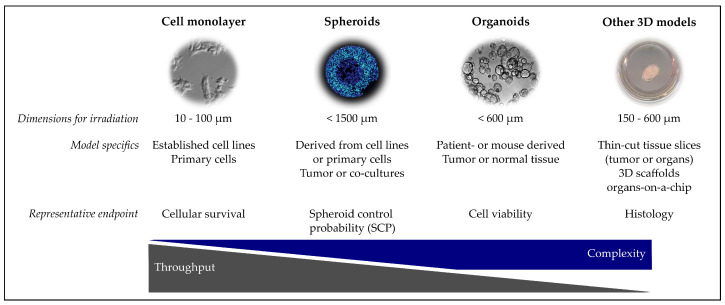
Overview of in vitro models for proton radiobiology experiments. Images (left to right): migrating human uveal melanoma cells [22], fluorescence staining of a tumor spheroid [72], human pancreatic cancer organoids (courtesy of Max Naumann), tumor slice culture [39]. Dimension refers to the required beam path length for irradiating the respective model.

**Figure 3 cancers-13-04216-f003:**
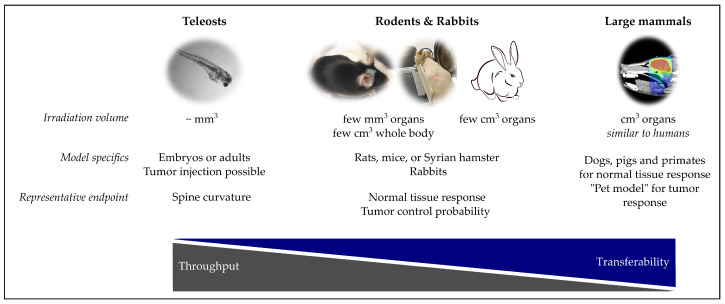
Overview of in vivo models for proton radiobiology experiments. Images (left to right): Irradiated zebrafish embryo [68], C57BL/6 mouse after proton brain irradiation [59], rat proton irradiation setup at Institut Curie [32], rabbit photon treatment plan of a mini-pig brain [69].

**Figure 4 cancers-13-04216-f004:**
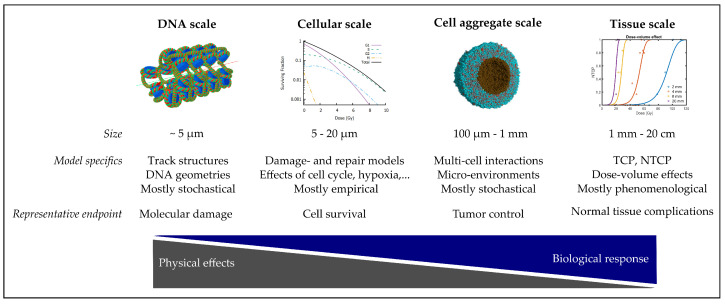
Overview of in silico models for proton radiobiology experiments. Images (left to right): chromatin fiber [280], surviving cell fraction in dependence on the cell cycle phase [281], simulated tumor spheroid 40 min post photon irradiation [70], NTCP model for increasing irradiation volumes (courtesy of E.Ba.).

**Table 1 cancers-13-04216-t001:** Literature overview of preclinical models for investigations of proton-specific research questions.

Model	RBE and LET	Normal Tissue Reaction	Molecular Responses	Tumor Biology	Combination Therapy	Personalized Medicine	Novel Radiation Treatment Modalities
2D cell culture	[20,37,83,90,91,92]	-	[20,21,24,75,76,77,78,79,80,81,82,93,94,95,96,97]	-	[99,100,101]	[104]	[36,78,84,85,86,87]
Spheroids, organoids	[38,40,111,138]	[38,143]	[38,133]	[135]	[40,133,136,137]	[38,111]	[72,139]
Other 3D models	[171]	[39]	-	[39]	-	-	[172]
Teleosts	[68,181]	[68,186]	-	[187,188,189]	-	[190,191]	[184]
Mice	[41,47,193,194,195,196,209,210,211,212]	[42,59,207,213,220,221,222]	[192,193]	[225,231,232]	[234]	-	[28,50,57,218,219,233,235]
Rats	[45,198,199,200,201,202,203]	[19,45,199,201,214,215,216,217,223,224]	-	-	-	[229]	[28,29,226,227,228]
Rabbits	-	[245,246]	-	[250,251,252,253,254,255]	-	-	-
Large mammals	-	[267,269,272,273]	-	[259]	-	-	[276]
In silico	[280,283,285,286,287,288,289,290,291,302,303,304,312,315,316,317,337,338]	[8,10,322,323,336,339,340]	[293,294,295,296,297,298,299]	[319,324]	[319]	[339]	-

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
