# Peer review of "Models for Translational Proton Radiobiology—From Bench to Bedside and Back"

_cancers, 2021, doi:10.3390/cancers13164216_

Round 1

Reviewer 1 Report

In the submitted manuscript entitled “Models for Translational Proton Radiobiology – From Bench to Bedside and Back”, Suckert and colleagues proposed an extensive review of the actual states of Proton radiobiology.

The manuscript is well written and addresses critical points for the field such as limitation of experimental beam time, choice of experimental models and systemic usage of a RBE of 1.1 despite the growing evidence that this fixed RBE is not optimal.

This is an impressive review of the literature and a great read for any new or actual scientists involved in proton radiobiology.

My only comment is about the lack of citation of the work done on FLASH proton radiotherapy in the rodent model which is actually the more advanced model tested. I would suggest the authors in order to be consistent with the other sections of the manuscript and to give the most up to date view of where the field stands in term of experimental set-up and capabilities to include a short section on the work already published on the topic:

Using experimental delivery sources:

10.1016/j.ijrobp.2019.10.049

10.1002/mp.14730

doi.org/10.1667/RADE-20-00068.1

Using a clinical gantry that led to an ongoing human feasibility study on the same delivery system

Preclinical study: 10.3390/cancers13051012

Feasibility study: https://www.clinicaltrials.gov/ct2/show/NCT04592887

Overall, I would strongly recommend this excellent review for publication as this is a much-needed overview and critical work pointing key elements for meaningful pre-clinical results towards translation to the clinic.

Author Response

We thank the reviewer for the recommendation and the constructive feedback on our manuscript. We agree that the FLASH studies conducted in rodents were missing and included the suggested references in the text (line 129, line 413, lines 461 ff), as well as in Table 1.

Reviewer 2 Report

This review summarized a wide spectrum of preclinical models related to proton therapy. The translational considerations were explained well in the physicist's and biologist's point of view. Various in vitro and in vivo systems were introduced. Since there have been few reviews that thoroughly described some translational knowledge to be used in particle therapy, this review article has its value to widen the preclinical principles of proton therapy. With regard to recent trends in clinical oncology, I recommend specific considerations of combined immuno-radiotherapy applying the proton therapy technique.

Author Response

We thank the reviewer for the positive feedback to our manuscript.

Combined immune-radiotherapy is indeed a very promising treatment option for clinical oncology. Therefore, we included studies on this topic with other radiation modalities in the “Rodents section” (lines 447 ff) and extended the conclusion (lines 652 ff) to point out this emerging research field. Furthermore, two references for immunoradiotherapy with other radiation modalities were added (ref 236 & 237).

Reviewer 3 Report

The topic of this review manuscript is extremely important and timely. Better understanding of the valuable features as well as limitations of proton radiotherapy is highly beneficial for the research and clinical communities. This review manuscript brings well organized and comprehensively written update on the current state of the various aspects of translational proton radiobiology. There are only couple of suggestions/questions from this reviewer to clarify certain aspects:

  1. Since cell culture is not the typical in vitro model and to distinguish it from “test tube” experimental models, it would be more correct to change title (page 4) “3. In vitro models” to “3. In vitro cell culture models”.
  2. In Figure 1 legend, it is not clear why parentheses are used: “Translational research in (proton) radiobiology…”

Author Response

We are grateful for the positive feedback to our manuscript. Thank you for pointing out the potentially misleading title, we changed the manuscript accordingly.

Regarding the parenthesis in Figure legend 1: In order to prevent copyright infringements, example images integrated into the figure were only taken out of open access publications. Unfortunately, we could not find suitable graphical material from the field of proton radiobiology for in silico and large mammals. Therefore, examples from photon radiobiology were used. To avoid misleading the reader, “proton” was put in parentheses.